# Journey of the tuberculosis patients in India from onset of symptom till one-year post-treatment

**Susmita Chatterjee**[1,2,3]*, **Palash Das**[1], **Aaron Shikhule**[2], **Radha Munje**[4], **Anna Vassall**[5]

**1** Research Department, George Institute for Global Health, New Delhi, India, **2** Department of Medicine, University of New South Wales, Sydney, New South Wales, Australia, **3** Prasanna School of Public Health, Manipal Academy of Higher Education, Manipal, Karnataka, India, **4** Department of Respiratory Medicine, Indira Gandhi Government Medical College, Nagpur, Maharashtra, India, **5** Department of Global Health and Development, London School of Hygiene and Tropical Medicine, London, United Kingdom

* schatterjee@georgeinstitute.org.in

**Data Availability Statement:** All data underlying the findings in this article is openly available and submitted with the manuscript as supporting file.

## Abstract

Historically, economic studies on tuberculosis estimated out-of-pocket expenses related to tuberculosis treatment and catastrophic cost, however, no study has yet been conducted to understand the post-treatment economic conditions of the tuberculosis patients in India. In this paper, we add to this body of knowledge by examining the experiences of the tuberculosis patients from the onset of symptoms till one-year post-treatment. 829 adult drug-susceptible tuberculosis patients from general population and from two high risk groups: urban slum dwellers and tea garden families were interviewed during February 2019 to February 2021 at their intensive and continuation phases of treatment and about one-year post-treatment using adapted World Health Organization tuberculosis patient cost survey instrument. Interviews covered socio-economic conditions, employment status, income, out-of-pocket expenses and time spent for outpatient visits, hospitalization, drug-pick up, medical follow-ups, additional food, coping strategies, treatment outcome, identification of post-treatment symptoms and treatment for post-treatment sequalae/recurrent cases. All costs were calculated in 2020 Indian rupee (INR) and converted into US dollar (US$) (1 US$ = INR 74.132). Total cost of tuberculosis treatment since the onset of symptom till one-year post-treatment ranged from US$359 (Standard Deviation (SD) 744) to US$413 (SD 500) of which 32%-44% of costs incurred in pre-treatment phase and 7% in post-treatment phase. 29%-43% study participants reported having outstanding loan with average amount ranged from US$103 to US$261 during the post-treatment period. 20%-28% participants borrowed during post-treatment period and 7%-16% sold/mortgaged personal belongings. Therefore, economic impact of tuberculosis persists way beyond treatment completion. Major reasons of continued hardship were costs associated with initial tuberculosis treatment, unemployment, and reduced income. Therefore, policy priorities to reduce treatment cost and to protect patients from the economic consequences of the disease by ensuring job security, additional food support, better management of direct benefit transfer and improving coverage through medical insurances need consideration.

**Funding:** This work was supported by the DBT/ Wellcome Trust India Alliance Clinical and Public Health Intermediate Fellowship [grant number IA/ CPHI/17/1/503339] awarded to SC. SC and PD received salary support from the funder. The funder had no role in study design, data collection and analysis, decision to publish, or preparation of the manuscript.

**Competing interests:** The authors have declared that no competing interests exist.

## Introduction

Tuberculosis (TB) is not only a global public health concern with estimated 10 million people suffered from the disease and 1.4 million people died in 2019 [1], but it also has societal consequences. TB is a contagious disease that requires long treatment and care and strongly associated with social stigma, poverty, illiteracy, unemployment, and catastrophic cost [2–7]. While much is known about the costs of TB treatment to households, in recent years there has been increasing emphasis on the long-term impact of TB treatment. Menzies et al (2021) estimated the health losses caused by global incidence of TB in 2019 (including the loss in the post-treatment period) and reported 122 million disability adjusted life years (DALYs) attributed to TB, of which, 58 million DALYs attributed to post-treatment sequalae [8]. Pulmonary TB patients may develop respiratory infections even after cure which may lead to greater morbidity and mortality. Further, TB survivors have a higher risk of disease recurrence. Therefore, the ongoing physical impact of the disease may have a long-term economic impact. A recent study conducted in Malawi examined whether the economic impact of the disease was fully recovered after one-year post-treatment. They concluded that many TB patients experienced limited recovery in income and employment with ongoing dissaving and schooling interruptions in the post- treatment period [9].

India has the highest TB burden in the world with an estimated incidence of 2.64 million in 2019 [10]. The government of India provides free diagnosis and treatment to all registered TB patients; however, studies reported high out-of-pocket expenses and catastrophic cost related to TB [11–15]. A recent literature review found that the mean total cost (direct and indirect) to the patients and households for drug-susceptible (DS) TB treatment in public health facilities was US$235 (Standard Deviation [SD] 210) at 2018 prices [16]. Further, the study showed that 7% to 32% of DS-TB patients in India faced catastrophic cost because of TB (defined as total cost $\geq$ 20% of total annual household income) [16]. While studies estimated the patient and household costs related to TB treatment in India, however, the patients for those studies were sampled from only one state/district covering limited geographical area [11–15]. Further, no study has yet been conducted to understand the post treatment economic conditions of the TB patients in India. Understanding the socioeconomic impact of TB patients during treatment and post-treatment will be crucial to develop strategies to reduce the burden and to improve the overall well-being of the TB survivors. In this paper, we add to this body of knowledge by examining the experiences of the TB patients in India for the full course of the disease and beyond: from the onset of symptoms till one year post treatment.

## Methods

The study design follows a cohort of total 1536 DS-TB patients among three groups: general population and two high-risk groups as identified in the National Strategic Plan for TB: 2017–2025 [17]: urban slum dwellers and patients from tea garden areas. The high-risk groups were chosen as the treatment seeking behaviour, costs of TB treatment and coping strategies among participants from high-risk groups could be different from general population and different protective measures may be required for them in the process of TB elimination in the country.

As per World Bank 2018 estimates, 35% of India's urban population live in slum areas [18]. Apart from having poor housing condition, sanitation, access to safe drinking water, the urban slum dwellers experience illegality and social exclusion, poorly regulated and ineffective health services [19].

India's tea industry is one of the largest private employers, and the people living in tea estates entirely rely on estates for employment and other services such as housing, water, health, education [20]. Because of very poor living condition and extremely low wage rate, the

families living in tea garden areas are at high-risk of any disease including TB. Their health service utilization pattern, treatment expenses, employment status in pre-, during and post TB period are unknown. Similar for urban slum dwellers. Hence, the main study aims to report the complete journey of the TB patients from general population as well as from tea garden and urban slum areas.

## Sampling strategy

For the costing study, national representativeness can be achieved by considering a limited number of states selected using an appropriately stratified sampling design. The World Health Organization Study on global AGEing and adult health (SAGE) sampling design was followed in this context [21]. As a first step, 29 states of India were stratified into six levels of development and six geographic locations. A composite development index was constructed using the following indicators at the state level: Infant mortality rate; female literacy rate; full immunization coverage rate; and per capita income. Principal component analysis technique was used to construct the composite index. Quantiles were then used to categorize the states into six levels of development (S1 Table). Based on the availability of study budget, Assam, Maharashtra, Tamil Nadu, and West Bengal were selected for this study as these states not only represent different levels of development but also cover different regions of the country. Apart from that, Assam, Tamil Nadu, and West Bengal have substantial land area for tea gardens, and Maharashtra, Tamil Nadu and West Bengal are among the top five states in terms of urban slum population in the country. TB patients from general population were drawn from all four states.

In the next stage of sampling, 3–5 districts from each state were selected purposively based on the dominance of the high-risk groups. From the sampled study districts, TB units (TUs— one TU covers 200,000 population (range 150,000–250,000) for rural and urban areas, 100,000 (range 75,000–150,000) in hilly, tribal, and difficult areas) were then identified which cater to the study high-risk groups. As all TUs cater to general population, there was no specific identification of TUs for covering patients from general population.

## Patient recruitment

In the next step, from all identified TUs, adult (18 years and above) TB patients who were at their intensive phase of treatment (DS-TB requires a minimum of six months of treatment, of which the first two months are called the intensive phase and the following four months the continuation phase) during the visit of the study team and gave written informed consent to participate in the study were interviewed.

## Sample size for each group for the main study

Number of DS-TB patients required in each group i.e., from general population, tea garden areas and urban slum dwellers was calculated to estimate the mean cost, if the resulting estimate is to fall within 7% of the true estimate with 95% confidence. The following formula was used to derive the number of DS-TB patients that must be sampled in each group

$$n = Z^2_{1-\alpha/2} \frac{\sigma^2}{(\varepsilon.\mu)^2} DEFF$$

where $\sigma$ is the population standard deviation, $\mu$ is the population mean, $\epsilon$ is the relative precision, $Z_{1-\frac{\alpha}{2}}$ is the $\left(1-\frac{\alpha}{2}\right)^{th}$ quantile of the standard normal distribution and DEEF is the design effect to account for the cluster sampling design. Design effect helps account for the clustering in costs borne by patients from the same TU. An earlier study conducted in Tamil Nadu state

in India [22], reported a mean cost of Indian Rupee (INR) 23,991 along with a standard deviation of INR 12,258 (converted in 2015 prices). Therefore, $\mu$ = INR 23,991 and $\sigma$ = INR 12,258 were assumed. For the resulting estimate to fall within 7% of the true estimate with 95% confidence, $\epsilon$ = 7% and $\alpha$ = 0.05, i.e. $Z_{1-\frac{\alpha}{2}} = 1.96$ were set. For most demographic health surveys, the design effect, which is the ratio of variances under cluster sampling and simple random sampling, comes out to be less than or equal to 2. Therefore, DEFF was set as 2. Based on these specifications, 410 DS-TB patients are required to estimate the mean cost within 7% of the true estimate with 95% confidence accounting for survey design. Considering a 10% loss to follow-up and another 10% non-response, the final sample size of TB patients required was 512 in each group. This implies that 1,536 DS-TB patients (512 patients in each group x three groups: general population, tea garden areas and slum dwellers) were required to be interviewed in total for the main study. 512 patients in each group were sampled from all four states except for patients in tea garden areas who were sampled from three states as the state Maharashtra does not have tea gardens.

## Present study participants

In this paper, we present the experience of 829 TB patients of whom 435 were from tea garden areas, 260 from general population and 134 from urban slum dwellers scattered across 9 districts, 52 TB units and 182 tea gardens as one-year post-treatment follow-up completed for these patients. This study participants were interviewed between February 2019 to February 2021 by trained researchers under close supervision using the adapted TB patient cost survey tool developed by the World Health Organization [23]. They were interviewed at their intensive phase (IP) (0–2 months) of treatment, end of continuation phase (CP) of treatment (5–6 months) and about one-year post-treatment.

IP interviews covered their socio-economic conditions and ownership of assets, employment status and patient and household income before onset of TB symptoms and at the time of interview, consumption expenditure of the household before TB, and out-of-pocket expenses (including consultation fees, medicines, laboratory / radiology tests, travel expenses, food and any other expenses) and time spent by the patients and the accompanied persons for outpatient visits, hospitalizations (out-of-pocket expenses included bed charges, consultation fees, medicines, tests, procedures such as biopsy / surgeries, travel expenses, additional food, accommodation and any other expenses) starting from onset of symptoms. Time spent by the patients and accompanied persons for TB drug-pick up / directly observed treatment (DOT), medical follow-ups were collected along with out-of-pocket expenses for travel, food during drug pick-up/DOT, medical follow-ups, additional food/nutritional supplements, and coping strategies (dissaving, borrowing, sold assets) during IP interviews. CP interviews covered time spent and out-of-pocket expenses related to TB treatment, employment and income status and coping strategies since the IP interviews. Post-treatment interviews covered treatment outcome, identification of post-treatment symptoms, expenses and time spent on outpatient visits and hospitalization related to post treatment sequalae, treatment cost of relapse cases, income and employment status, and socioeconomic consequences such as outstanding loan (i.e., the loan the study participants took for TB diagnosis/treatment during pre- and treatment period but could not repay at the time of post-treatment interview), borrowing/sale/mortgage of belongings and others (if any).

## Impact of COVID-19 on the present study

In India, to curb the spread of the coronavirus infection, there was a nationwide lockdown from March 25, 2020, till June 8, 2020. The present study participants were severely impacted

by the restrictive measures and there was huge loss of employment and income among the study participants [24]. To understand the impact of TB only, participants were asked to report employment status, individual and household incomes for the month of February 2020 (i.e., the month before the nationwide lockdown started). Further, during the interviews, they were reminded to report about coping strategies used only for TB disease, not for managing the crisis during COVID-19. Reported outstanding loan amounts in the post-treatment period were cross checked with the amounts borrowed during treatment period for consistency.

## Cost calculation methods

Pre-treatment cost was defined as the costs (both direct and indirect costs, i.e., actual money spent and time cost) incurred from the onset of TB symptoms till the date of treatment initiation. Total treatment cost was the sum of direct and indirect cost incurred during IP and CP of treatment. For patients whose treatment were extended, costs were calculated till the completion of treatment. For the defaulters, TB drug pick up costs were calculated till the date of last drug pick up. Time costs of patient and accompanied persons were calculated using the 'human capital approach' [25] where hours spent for each activity related to TB treatment was multiplied by the minimum hourly wage rate of the respective states [26]. Catastrophic cost was calculated as proportion of total TB treatment cost of pre-TB annual household income. Total TB treatment cost, exceeding a given threshold (20%) of the household's annual pre-TB income, was considered as catastrophe [23].

Post TB infections present similar features as pulmonary TB such as weight loss, chronic productive cough, coughing up blood, difficulty in breathing and fatigue [27]. Study participants were asked if they experienced any such symptoms during post-treatment period and whether availed health care for managing those symptoms. Costs of outpatient visits, hospitalization and treatment cost for the relapse cases were calculated using same methodology discussed earlier. All costs were calculated in 2020 Indian rupees (INR) and converted into US dollars (US$) using average exchange rate of 2020 (1 US$ = INR 74.132).

To determine the relationships of post-treatment financial hardship with several predictor variables, we ran two multiple linear regressions for each group. After testing for several models, we selected linear regression model as the best fit. In one regression, outstanding loan amount in the post-treatment period was the dependent variable and direct cost of initial TB treatment, post-treatment household income, treatment cost in the post-treatment period, age, gender, and wealth index were the independent variables. Wealth index is a composite measure of a household's cumulative living standard. During the IP interviews, data were collected on household type, materials used for house construction, drinking water and sanitation facilities, usage of cooking fuel, and ownership of assets such as television, refrigerator, mobile phone, computer, internet connection, motorcycle, car, jeep etc. Variables were chosen from national demographic and health surveys in India. Principal component analysis was used to create the wealth index for each study participant.

In another regression model, amount of borrowing and sale/mortgage of belongings (i.e., the amount received combining all coping strategies) in the post-treatment period was the dependent variable and post-treatment household income, treatment cost in the post-treatment period, outstanding loan amount in the post-treatment period, age, gender, and wealth index were the explanatory variables. The choice of the independent variables was motivated by the study in Malawi [9] and from the authors' understanding of the data. Descriptive statistics of the predictor variables are presented in S2 Table. Preliminary analyses were performed to ensure there was no violation of the assumptions of multicollinearity. The fitted line plots indicating the linear relationship between dependent and independent variables are given in S1–S6 Figs.

### Ethics approval and consent to participate

The study was approved by the Institutional Ethics Committee of the George Institute for Global Health (014/2018). Written informed consent was obtained from each study participant before starting the interview.

## Results

### Characteristics of the study participants

A total of 829 DS-TB patients were interviewed in IP, however, one year post treatment, 705 participants were interviewed (about 15% loss to follow-up). The reasons of loss to follow up are given in Table 1. Average recall period from onset of symptoms to treatment initiation ranged from 52 days (SD 57) to 54 days (SD 55) while the recall period from treatment initiation to IP interview ranged from 29 days (SD 17) to 36 days (SD 13). Average recall period for CP interviews was between 120 days (SD 15) and 162 days (SD 35) and for post-treatment interviews, it ranged from 272 days (SD 75) to 314 days (SD 91).

Characteristics of the study participants interviewed in IP and post treatment were similar (Table 2). Most of the participants were male and were in younger age group, 18–34 years. About 40% participants in tea garden areas never attended school, one third in general population attended primary school and 47% in slum areas completed higher secondary education. Average household monthly income before the study participant had TB was between $100-$199 for 38% participants in general population and 45% among slum dwellers while it was less than $100 for 53% of the participants from tea garden areas (Table 2). Most of the study participants had pulmonary bacteriologically confirmed TB while 25% to 35% had extrapulmonary TB with cervical lymph node and plural effusion were the most common forms.

**Table 1. Reasons of loss to follow up.**

| | Intensive phase—N | Continuation phase–N | Reasons–N (%) | Post- treatment follow-up–N | Reasons–N (%) |
|---|---|---|---|---|---|
| General population | 260 | 234 | Death– 10 (3.85%) | 221 | Death– 3 (1.28%) |
| | | | Refusal– 8 (3.08%) | | Refusal– 6 (2.56%) |
| | | | No trace– 7 (2.69%) | | No trace– 4 (1.71%) |
| | | | Migrate out– 1 (0.38%) | | Total– 13 (5.55%) |
| | | | Total– 26 (10%) | | |
| Tea garden families | 435 | 396 | Death– 22 (5.06%) | 379 | Death– 10 (2.64%) |
| | | | Regimen change (shifted to DR)– 9 (2.07%) | | No trace– 6 (1.58%) |
| | | | No trace– 5 (1.15%) | | Refusal– 1 (0.26%) |
| | | | Migrate out– 2 (0.46%) | | Total– 17 (4.48%) |
| | | | Cancelled– 1 (0.23%) | | |
| | | | Total– 39 (8.97%) | | |
| Urban slum dwellers | 134 | 127 | Death– 2 (1.49%) | 105 | Death– 5 (3.94%) |
| | | | No trace– 1 (0.75%) | | No trace– 10 (7.81%) |
| | | | | | Refusal– 7 (5.47%) |
| | | | Refusal– 3 (2.24%) | | Total– 22 (17.32%) |
| | | | Regimen change (shifted to DR)– 1 (0.75%) | | |
| | | | Total– 7 (5.22%) | | |

Note: DR–Drug resistant TB

**Table 2. Characteristics of the study participants.**

| | General population | | Tea garden areas | | Urban slum dwellers | |
|---|---|---|---|---|---|---|
| **Age** | **Intensive phase (N = 260)** | **Post-treatment follow-up (N = 221)** | **Intensive phase (N = 435)** | **Post-treatment follow-up (N = 379)** | **Intensive phase (N = 134)** | **Post-treatment follow-up (N = 105)** |
| 18–24 years | 61 (23.5%) | 55 (24.9%) | 115 (26.4%) | 109 (28.8%) | 47 (35.1%) | 41 (39.0%) |
| 25–34 years | 65 (25.0%) | 58 (26.2%) | 134 (30.8%) | 112 (29.6%) | 30 (22.4%) | 24 (23.0%) |
| 35–44 years | 57 (21.9%) | 50 (22.6%) | 91 (20.9%) | 80 (21.1%) | 19 (14.2%) | 14 (13.3%) |
| 45–54 years | 43 (16.5%) | 33 (14.9%) | 48 (11.0%) | 38 (10.0%) | 22 (16.4%) | 14 (13.3%) |
| 55–64 years | 20 (7.7%) | 16 (7.2%) | 37 (8.5%) | 32 (8.4%) | 10 (7.5%) | 8 (7.6%) |
| 65 years and above | 14 (5.4%) | 9 (4.1%) | 10 (2.3%) | 8 (2.1%) | 6 (4.5%) | 4 (3.8%) |
| **Gender** | | | | | | |
| Male | 179 (68.8%) | 153 (69.2%) | 258 (59.3%) | 218 (57.5%) | 71 (53.0%) | 55 (52.4%) |
| Female | 81 (31.2%) | 68 (30.8%) | 177 (40.7%) | 161 (42.5%) | 63 (47.0%) | 50 (47.6%) |
| **Education** | | | | | | |
| Not attended school | 51 (19.6%) | 44 (19.9%) | 173 (39.8%) | 145 (38.3%) | 34 (25.4%) | 24 (22.9%) |
| Primary education | 75 (28.8%) | 63 (28.5%) | 148 (34.0%) | 128 (33.8%) | 15 (11.2%) | 13 (12.4%) |
| Secondary and above | 134 (51.6%) | 114 (51.6%) | 114 (26.2%) | 106 (28.0%) | 85 (63.4%) | 68 (64.8%) |
| **Before TB monthly household income** | | | | | | |
| No income | 2 (0.8%) | 0 (0.00%) | 4 (0.9%) | 2 (0.5%) | 2 (1.5%) | 1 (0.9%) |
| Less than $100 | 57 (21.9%) | 53 (24.0%) | 229 (52.6%) | 201 (53.0%) | 34 (25.4%) | 23 (21.9%) |
| $100 - $199 | 98 (37.7%) | 86 (39.0%) | 164 (37.7%) | 140 (36.9%) | 60 (44.8%) | 46 (43.8%) |
| $200 - $299 | 47 (18.1%) | 38 (17.2%) | 28 (6.4%) | 26 (6.9%) | 30 (22.4%) | 28 (26.7%) |
| $300 - $399 | 18 (6.9%) | 18 (8.1%) | 6 (1.4%) | 6 (1.6%) | 7 (5.2%) | 7 (6.7%) |
| $400 - $499 | 13 (5.0%) | 8 (3.6%) | 0 (0.0%) | 0 (0.0%) | 0 (0.0%) | 0 (0.0%) |
| $500 and above | 25 (9.6%) | 18 (8.1%) | 4 (0.9%) | 4 (1.1%) | 1 (0.7%) | 0 (0.0%) |
| **Type of TB** | | | | | | |
| Pulmonary, bacteriologically confirmed | 145 (55.8%) | 125 (56.6%) | 262 (60.2%) | 221 (58.3%) | 89 (66.4%) | 66 (62.9%) |
| Pulmonary, bacteriologically unconfirmed | 26 (10.0%) | 19 (8.6%) | 63 (14.5%) | 55 (14.5%) | 5 (3.7%) | 5 (4.7%) |
| Extra pulmonary | 89 (34.2%) | 77 (34.8%) | 110 (25.3%) | 103 (27.2%) | 40 (29.9%) | 34 (32.4%) |
| **Residential status** | | | | | | |
| Urban | 112 (43.1%) | 91 (41.2%) | 0 (0.0%) | 0 (0.0%) | 105 (100.0%) | 134 (100.0%) |
| Rural | 148 (56.9%) | 130 (58.8%) | 435 (100.0%) | 379 (100.0%) | 0 (0.0%) | 0 (0.0%) |

Kruskal Wallis test after Bonferroni adjustment showed all variables except type of TB in intensive phase were significantly different among three groups.

## Employment and income status of the study participants

Changes in employment and income status of the study participants in different phases of treatment and post-treatment period are presented in Figs 1 and 2 respectively. Results are presented for 705 participants who were followed up till the post-treatment period.

Before TB, approximately 6% (urban slum dwellers) to 10% (tea garden areas) of the study participants were unemployed, however, in IP, the percentage of unemployment ranged from 32% among slum dwellers to 64% among participants in tea garden areas implying that TB had serious impact on participants' employment (Fig 1). In CP, the situation improved, and unemployment rate ranged from 27%-47%. About one-year post-treatment, however, all study participants could not return to their pre-TB condition. Unemployment rate in post-TB period ranged from 21%-31% as compared to 6%-10% in the pre-TB condition.

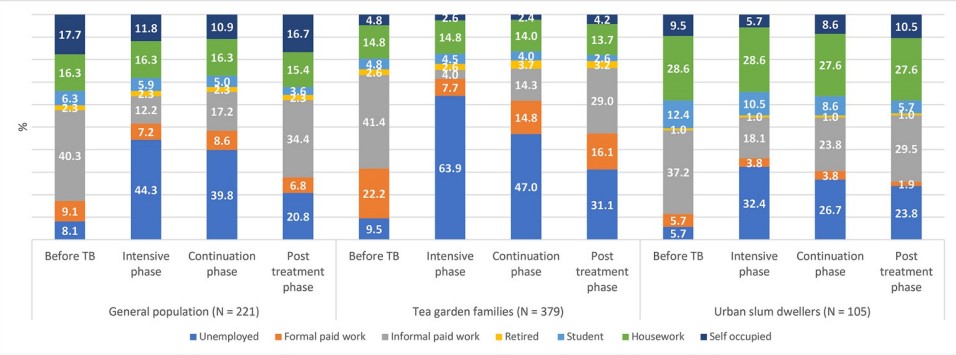

**Fig 1. Employment status of the sampled patients at different phases.**

Before TB, no households of study participants in general population had zero income (Fig 2). However, in IP, about 13% households had zero income probably indicating that those patients were the sole earning members of the family. In post-treatment period, 3% households had zero income. Proportion of households in higher income groups gradually moved towards lower income group during IP indicating that patients were unable to work fully. In the post-treatment period, proportion improved but was lower as compared to pre-TB proportion. Similar trends were observed for other two groups (Fig 2).

## Heath seeking behaviour of the study participants at the pre-treatment phase

Equal proportion of participants in general population first visited private facilities (33%) and government facilities (33%) after the onset of suggestive TB symptoms followed by 23% in drug stores and 5% to unqualified practitioners. The most preferred facility for seeking treatment for the participants in tea garden areas was the tea garden hospitals or dispensaries (42%) followed by government facilities (28%), drug stores (13%) and private facilities (12%). For participants in urban slum areas, first preference was drug stores (26%) followed by private facilities (25%). 19% patients first visited the unqualified practitioners and a similar proportion visited government facilities after the onset of TB symptoms. Hospitalization rate ranged from 16% (participants in slum areas) to 29% (tea garden areas). 52% of those hospitalized in tea garden areas were admitted in tea garden hospitals followed by 35% in government hospitals. For other two groups, 64%-76% hospitalizations were in public. 20% study participants in

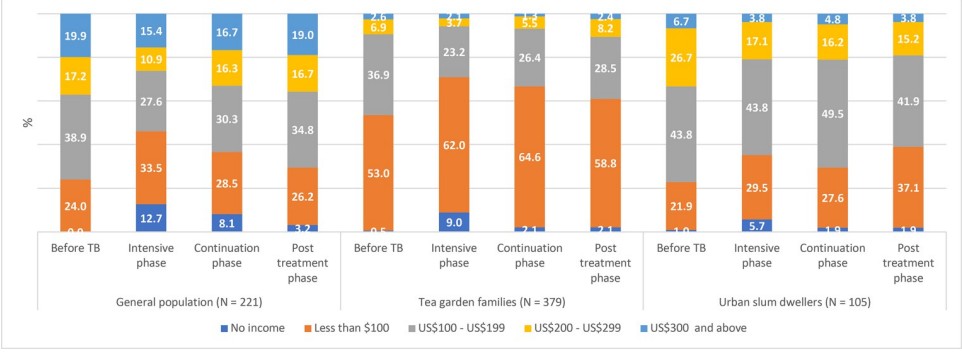

**Fig 2. Income status of the households of sampled TB patients at different phases.**

general population and 34% patients in tea garden areas reported having health insurance while only 2% participants among urban slum dwellers had health insurance. Only couple of hospitalized patients reported that they got benefit from their health insurances.

Average time from the onset of TB symptoms till the start of TB treatment was 8 weeks (SD 8) for participants in general population and slum dwellers (with a range of 1 week to 70 weeks) and for participants in tea garden areas, 7 weeks (SD 8) (ranging from 1 week to 60 weeks). During that time, study participants among slum dwellers made an average of 12 (SD 8) visits at different facilities (ranged from 1 to 52) while participants among general population made 9 (SD 5) visits (range: 1 to 31) and participants from tea garden areas 8 (SD 4) visits (range: 1 to 25).

## Post-treatment sequalae and treatment seeking behaviour

After about one-year post-treatment, participants in all categories reported multiple symptoms similar as pulmonary TB. 29% participants in general population (63/221) reported lack of energy, 25% suffering from shortness of breath, 14% was coughing during the interview and 14% having chest pain. 16% participants visited different health facilities for treatment of these symptoms and 46% among them went to the public facilities first while 29% went to the private facilities.

Symptoms were similar for participants in tea garden areas and urban slum dwellers. 30% participants in tea garden areas and 34% in slum areas had at least one symptom. 18% participants in tea garden areas sought treatment for these symptoms and majority (43%) went to the tea garden hospitals first for treatment and 31% went to public health facilities. Preference for health facility changed for urban slum dwellers when compared with pre-treatment phase, 53% of those who made visits during post-treatment period went to government health facilities first instead of visiting drug stores or private facilities. Hospitalization rate was much lower in post-treatment period as compared to pre-treatment, only two participants each among general population and urban slum dwellers were hospitalized while three were hospitalized among tea garden area participants.

## Costs of TB treatment

Costs incurred by study participants during TB treatment and for post-treatment sequalae / relapse cases are presented in Table 3. Total cost of TB treatment since the onset of TB symptom till one-year post-treatment for the general population was US$397 (SD 490) of which 44% was incurred in pre-treatment phase, 17% in IP, 33% in CP and 6% in post-treatment phase. In all phases, direct cost was the major contributor ranging from 63% of total costs of pre- and post-treatment phases to 84% of IP total cost. The reasons of high direct costs during IP and CP were purchase of additional nutritional food and supplements while in pre- and post-treatment phases, major direct cost was for outpatient visits.

For participants in tea garden areas, total treatment cost (starting from TB symptom till one-year post-treatment) was US$359 (SD 744) with 44% cost incurred in CP, 40% in pre-treatment, 11% in IP and 6% in post-treatment phase. Indirect cost dominated in all phases except in IP and it ranged from 63% to 84% of total costs of those phases respectively. Time cost for hospitalization was the reason of high indirect cost.

TB treatment cost for participants in urban slum areas was US$413 (SD 500). 40% of total cost was incurred in CP followed by 32% in pre-treatment phase, 21% in IP and 7% in post-treatment period. Direct cost was the major cost in all phases except in post-treatment ranging from 52% of total cost incurred in pre-treatment phase to 74% in IP. In pre-treatment phase, costs incurred for outpatient visits were the major direct cost, while in IP and CP, costs

**Table 3.  Costs of tuberculosis treatment (US$ 2020).**

| | General population (N = 221) | | | | | | | | | | | |
| --- | --- | --- | --- | --- | --- | --- | --- | --- | --- | --- | --- | --- |
| | Pre-treatment phase | | | Intensive phase | | | Continuation phase | | | Post-treatment phase | | |
| | Direct cost | Indirect cost | Total cost | Direct cost | Indirect cost | Total cost | Direct cost | Indirect cost | Total cost | Direct cost | Indirect cost | Total cost |
| Outpatient visit | 4944.13 | 1827.49 | 6771.62 | 118.74 | 38.59 | 157.33 | 1055.15 | 455.62 | 1510.77 | 821.59 | 330.47 | 1152.07 |
| Hospitalization | 3239.41 | 3051.95 | 6291.36 | 0.00 | 0.00 | 0.00 | 1576.70 | 295.57 | 1872.26 | 9.23 | 206.65 | 215.88 |
| Drug pick up + DOT | 0.00 | 0.00 | 0.00 | 332.05 | 784.54 | 1116.59 | 653.62 | 1132.94 | 1786.56 | 60.27 | 110.14 | 170.42 |
| Additional food | 0.00 | 0.00 | 0.00 | 3724.91 | 0.00 | 3724.91 | 4544.12 | 0.00 | 4544.12 | 216.96 | 0.00 | 216.96 |
| | Tea Garden Families (N = 379) | | | | | | | | | | | |
| | Pre-treatment phase | | | Intensive phase | | | Continuation phase | | | Post-treatment phase | | |
| | Direct cost | Indirect cost | Total cost | Direct cost | Indirect cost | Total cost | Direct cost | Indirect cost | Total cost | Direct cost | Indirect cost | Total cost |
| Outpatient visit | 1815.22 | 1317.71 | 3132.94 | 13.54 | 15.98 | 29.52 | 346.26 | 291.68 | 637.94 | 144.00 | 98.38 | 242.38 |
| Hospitalization | 749.66 | 6860.70 | 7610.36 | 0.00 | 0.00 | 0.00 | 248.56 | 6519.21 | 6767.77 | 158.43 | 485.78 | 644.21 |
| Drug pick up + DOT | 0.00 | 0.00 | 0.00 | 150.09 | 1125.57 | 1275.67 | 189.39 | 1940.35 | 2129.74 | 28.38 | 175.05 | 203.43 |
| Additional food | 0.00 | 0.00 | 0.00 | 1560.18 | 0.00 | 1560.18 | 2134.74 | 0.00 | 2134.74 | 187.46 | 0.00 | 187.46 |
| | Urban slum dwellers (N = 105) | | | | | | | | | | | |
| | Pre-treatment phase | | | Intensive phase | | | Continuation phase | | | Post-treatment phase | | |
| | Direct cost | Indirect cost | Total cost | Direct cost | Indirect cost | Total cost | Direct cost | Indirect cost | Total cost | Direct cost | Indirect cost | Total cost |
| Outpatient visit | 2995.04 | 1552.52 | 4547.56 | 95.17 | 64.17 | 159.34 | 291.16 | 427.46 | 718.62 | 142.10 | 89.23 | 231.34 |
| Hospitalization | 2091.79 | 3140.38 | 5232.17 | 0.00 | 0.00 | 0.00 | 2160.11 | 1027.30 | 3187.42 | 22.29 | 1586.25 | 1608.54 |
| Drug pick up + DOT | 0.00 | 0.00 | 0.00 | 463.88 | 1649.63 | 2113.51 | 876.37 | 2304.76 | 3181.13 | 36.57 | 122.78 | 159.35 |
| Additional food | 0.00 | 0.00 | 0.00 | 4273.22 | 0.00 | 4273.22 | 5203.28 | 0.00 | 5203.28 | 18.82 | 0.00 | 18.82 |

DOT–Directly observed treatment; 1 US$ = INR 74.132

incurred for buying additional food were the major contributors. Indirect cost contributed 89% of total post-treatment phase cost and time cost for hospitalization was the major cost contributor. Kruskal Wallis test after Bonferroni adjustment showed significant difference in treatment cost among three groups (p<0.001).

## Catastrophic cost by wealth quintile

35% patients from general population, 38% among urban slum dwellers and 40% among tea garden areas faced catastrophic cost because of TB. The proportions remain the same using pre-TB household expenditure as denominator of catastrophic cost. Among patients who faced catastrophe in general population, the cost burden was the highest for the poorest quintile (48%) (Fig 3), however, for the other two groups, cost burden was the highest for the richest quintile.

## Treatment outcome

13 out of 260 study participants in general population died during the follow up period, majority deaths occurred during treatment phase. Similar proportion of participants in slum areas died, however, majority death occurred after completing their treatment. Death rate was higher for participants in tea garden areas (about 7%) and majority death occurred during IP. (9/260) participants in general population and (7/435) in tea garden areas were defaulters,

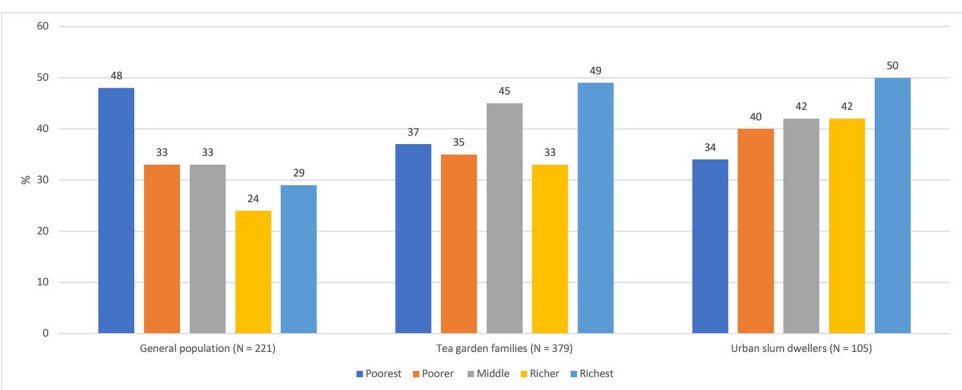

**Fig 3. Proportion of patients with catastrophic cost by wealth quintile.**

there were no defaulters among participants in urban slum areas. About 4% participants both in general population (8/221) and tea garden areas (16/379) had TB again after completing their treatment whereas only two participants had relapse cases in slum areas.

## Coping strategies during treatment and post-treatment period

High rate of unemployment during TB treatment along with expenses incurred for treatment forced study participants to borrow, sell/mortgage of assets and withdraw money from financial institutions. Details of coping strategies are presented in Table 4. Proportion of borrowing and sale during IP was the highest among general population while a significant proportion of participants in tea garden areas (54%) withdrawn from savings during IP. The borrowing and sale were the highest for them in CP. Types of items sold/mortgaged differed among the groups, while the participants from general population sold/mortgaged gold and jewellery (42%) followed by livestock (40%), land (17%) during treatment period, participants from tea garden areas mostly sold/mortgaged livestock (45%) followed by land (21%), farm produce (13%) and household items (11%). For the participants in slum areas, the most common item available for sell/mortgage was gold or jewellery (87%) followed by transport/vehicle, household items and business property (4% each).

During post-treatment period, 29% to 43% study participants reported having outstanding loan with average amount ranged from US$103 to US$261 (Table 4) and the proportion having outstanding loan was significantly different among three groups (p<0.001). 20%-28% participants borrowed during post-treatment period and 7%-16% sold/mortgaged household items. Like treatment period, gold/jewellery was the most common item for sale/mortgage for the general population (38%) followed by livestock (34%) and land (17%). Items for sale/mortgage for participants from tea garden areas were also similar as during treatment period, most common item being livestock (65%), followed by farm produce (17%) and land (13%). For participants from urban slum areas, mostly used item for sell/mortgage was gold/jewellery (57%) followed by household items (29%) and transport/vehicle (14%).

## Coping strategies by wealth quintile

Even though catastrophic cost was the highest among the richest quintiles for tea garden families and urban slum dwellers, highest proportion of participants in poorest quintiles in both groups used various coping strategies both during treatment and post-treatment (Table 5). Proportion of participants with outstanding loan in the post-treatment period was also highest among the poorest quintiles in each group.

**Table 4. Coping strategies during treatment and post-treatment period.**

| | General population (N = 221) | | | | | |
|---|---|---|---|---|---|---|
| | Intensive phase | | Continuation phase | | Post-treatment phase | |
| | Number | Amount (US$) | Number | Amount (US$) | Number | Amount (US$) |
| Borrowing | 104 (47.06%) | 195.00 | 63 (28.51%) | 183.21 | 58 (26.24%) | 195.48 |
| Sell / mortgage | 25 (11.31%) | 123.89 | 28 (12.67%) | 254.71 | 29 (13.12%) | 360.55 |
| Savings withdrawn | 46 (20.81%) | 267.97 | - - - | - - - | - - - | - - - |
| Outstanding loan | - - - | - - - | - - - | - - - | 95 (42.99%) | 261.12 |
| | Tea garden families (N = 379) | | | | | |
| | Intensive phase | | Continuation phase | | Post-treatment phase | |
| | Number | Amount (US$) | Number | Amount (US$) | Number | Amount (US$) |
| Borrowing | 113 (29.82%) | 101.70 | 93 (54.54%) | 64.57 | 74 (19.53%) | 89.35 |
| Sell / mortgage | 41 (10.82%) | 112.76 | 62 (16.36%) | 83.11 | 60 (15.83%) | 82.19 |
| Savings withdrawn | 93 (54.54%) | 112.04 | - - - | - - - | - - - | - - - |
| Outstanding loan | - - - | - - - | - - - | - - - | 110 (29.02%) | 102.87 |
| | Urban slum dwellers (N = 105) | | | | | |
| | Intensive phase | | Continuation phase | | Post-treatment phase | |
| | Number | Amount (US$) | Number | Amount (US$) | Number | Amount (US$) |
| Borrowing | 44 (41.90%) | 145.35 | 33 (31.43%) | 151.20 | 29 (27.62%) | 137.69 |
| Sell / mortgage | 10 (9.52%) | 124.78 | 13 (12.38%) | 140.40 | 7 (6.67%) | 136.82 |
| Savings withdrawn | 22 (20.95%) | 205.96 | - - - | - - - | - - - | - - - |
| Outstanding loan | - - - | - - - | - - - | - - - | 43 (40.95%) | 202.40 |

Note: savings withdrawn was mostly during the intensive phase, hence, the total amount was reported in intensive phase.

**Table 5. Proportion of households used coping strategies by wealth quintile.**

| | General population (N = 221) | | | |
|---|---|---|---|---|
| Wealth quintile | Borrowing (treatment) | Savings withdrawn (treatment) | Borrowing (post-treatment) | Outstanding loan (post-treatment) |
| Poorest | 75 | 16 | 48 | 59 |
| Poorer | 60 | 14 | 36 | 43 |
| Middle | 61 | 24 | 29 | 35 |
| Richer | 55 | 36 | 21 | 36 |
| Richest | 24 | 12 | 6 | 24 |
| | Tea garden families (N = 379) | | | |
| | Borrowing (Treatment) | Savings withdrawn (treatment) | Borrowing (post-treatment) | Outstanding loan (post-treatment) |
| Poorest | 50 | 23 | 36 | 33 |
| Poorer | 57 | 20 | 32 | 25 |
| Middle | 57 | 29 | 30 | 27 |
| Richer | 56 | 18 | 24 | 30 |
| Richest | 45 | 34 | 31 | 30 |
| | Urban slum dwellers (N = 105) | | | |
| | Borrowing (Treatment) | Savings withdrawn (treatment) | Borrowing (post-treatment) | Outstanding loan (post-treatment) |
| Poorest | 60 | 20 | 37 | 54 |
| Poorer | 74 | 23 | 29 | 34 |
| Middle | 58 | 32 | 26 | 42 |
| Richer | 75 | 0 | 25 | 33 |
| Richest | 25 | 25 | 25 | 25 |

**Table 6. Other economic consequences on sampled tuberculosis patients.**

| | General population (N = 221) | | Tea garden families (N = 379) | | Urban slum dwellers (N = 105) | |
|---|---|---|---|---|---|---|
| **Had to do the activities** | During treatment | Post- treatment | During treatment | Post- treatment | During treatment | Post- treatment |
| Cut down consumption level for other family members | 25 (11.31%) | 29 (13.12%) | 35 (9.23%) | 34 (8.97%) | 25 (23.81%) | 20 (19.05%) |
| Other household members started working | 5 (2.26%) | 1 (0.45%) | 4 (1.06%) | 0 (0.00%) | 2 (1.90%) | 0 (0.00%) |
| Withdrawn children from school / tuition | 1 (0.45%) | 1 (0.45%) | 4 (1.06%) | 5 (1.32%) | 1 (0.95%) | 1 (0.95%) |
| Run up account in grocery shop | 19 (8.60%) | 13 (5.88%) | 45 (11.87%) | 36 (9.50%) | 8 (7.62%) | 4 (3.81%) |
| Used multiple options | 32 (14.48%) | 29 (13.12%) | 30 (7.92%) | 14 (3.69%) | 18 (17.14%) | 11 (10.48%) |
| Did not use any strategy | 126 (57.01%) | 147 (66.52%) | 252 (66.49%) | 286 (75.46%) | 50 (47.62%) | 67 (63.81%) |
| Don't know | 13 (5.88%) | 1 (0.45%) | 9 (2.37%) | 4 (1.06%) | 1 (0.95%) | 2 (1.90%) |
| **Unable to do the following activities** | | | | | | |
| Could not pay electricity / mobile / gas / cable bills | 37 (16.74%) | 32 (14.48%) | 62 (16.36%) | 50 (13.19%) | 12 (11.43%) | 12 (11.43%) |
| Could not pay tuition fees | 5 (2.26%) | 0 (0.00%) | 4 (1.06%) | 1 (0.26%) | 2 (1.90%) | 2 (1.90%) |
| Could not pay house rent | 1 (0.45%) | 0 (0.00%) | 0 (0.00%) | 0 (0.00%) | 5 (4.76%) | 2 (1.90%) |
| Could not contribute to family / social events | 9 (4.07%) | 18 (8.14%) | 20 (5.28%) | 51 (13.46%) | 2 (1.90%) | 7 (6.67%) |
| Others | 3 (1.36%) | 1 (0.45%) | 1 (0.26%) | 1 (0.26%) | 3 (2.86%) | 0 (0.00%) |
| Used multiple options | 51 (23.08%) | 37 (16.74%) | 53 (13.98%) | 70 (18.47%) | 27 (25.71%) | 20 (19.05%) |
| Did not use any option | 110 (49.77%) | 131 (59.28%) | 230 (60.69%) | 198 (52.24%) | 52 (49.52%) | 60 (57.14%) |
| Don't know | 5 (2.26%) | 2 (0.90%) | 9 (2.37%) | 8 (2.11%) | 2 (1.90%) | 2 (1.90%) |

## Other economic consequences

Apart from borrowing, selling, and withdrawing money from banks / other financial institutions, patients also had other economic consequences such as cut down consumption for other family members, run up account in grocery shops, unable to pay electricity / mobile / gas / cable bills, could not contribute to family / social events not only during treatment period but also during post-treatment period (Table 6).

## Predictors of post-treatment financial hardship

We present the regression coefficients with all explanatory variables for each group in Table 7. We found that total direct cost of TB treatment (including direct cost incurred in pre-treatment phase) was significantly associated with outstanding loan amount in the post-treatment period for all groups. On the other hand, outstanding loan amount was significantly associated with borrowing / sale / mortgage in the post-treatment period implying that those who had outstanding loan, had to use various coping strategies in the post-treatment period. Treatment cost for post-treatment sequalae and relapse cases also forced the study participants in tea garden areas for distress financing in the post-treatment period. There were no association of financial hardship with socio-economic or demographic characteristics (except age which was marginally associated with dissaving for participants among general population and tea garden areas) implying that all categories of patients had distress financing in the post-treatment period. Dropping the socio-economic and demographic variables from the regression models from all groups did not affect the overall goodness-of-fit of the models.

## Discussion

The present study reports the journey of 705 TB patients in India from the onset of TB symptoms till one-year post-treatment. 829 patients were interviewed in IP however because of loss to follow up, post-treatment follow up interviews were completed with 705 TB patients. To the best of the authors' knowledge, this is the first study in India that examined the economic

**Table 7. Results of regression analysis of amount of outstanding loan and amount of borrowing /sale /mortgage of personal belongings during the post-treatment period.**

**Dependent variable: Amount of outstanding loan in the post-treatment period**

| Explanatory variables | Regression coefficients (95% CI) | | |
| --- | --- | --- | --- |
| | General population (N = 220) | Tea garden families (N = 379) | Urban slum dwellers (N = 104) |
| Direct cost of tuberculosis treatment | 0.183 (0.070, 0.296)*** | 0.137 (0.058, 0.217)*** | 0.150 (0.027, 0.273)* |
| Post-treatment household income | -0.079 (-0.221, -0.064) | -0.062 (-0.186, 0.063) | 0.039 (-0.230, 0.376) |
| Age | 203.841 (-44.174, 363.509)* | -9.741 (-66.391, 46.909) | 57.112 (-96.780, 221.003) |
| Gender (male) | 2007.636 (-2678.477, 6693.749) | 506.594 (-934.298, 1947.487) | -3610.117 (-7988.63, 768.398) |
| Cost of treatment in the post-treatment period | -0.055 (-0.220, 0.110) | 0.217 (0.139, 0.295)*** | 0.031 (-0.103, 0.166) |
| Wealth index | -586.973 (-2129.541, 955.595) | 382.463 (-119.916, 884.843) | -693.006 (-2114.973, 728.962) |
| Adjusted $R^2$ | 0.053 | 0.097 | 0.064 |

**Dependent variable: Amount of borrowing and amount received from sale / mortgage of personal belongings in the post-treatment period**

| Explanatory variables | Regression coefficients (95% CI) | | |
| --- | --- | --- | --- |
| | General population (N = 220) | Tea garden families (N = 379) | Urban slum dwellers (N = 105) |
| Cost of treatment in the post-treatment period | 0.141 (-0.272, 0.555) | 0.145 (0.086, 0.204)*** | 0.008 (-0.095, 0.111) |
| Post-treatment household income | -0.069 (-0.181, 0.042) | -0.043 (-0.133, .047) | -0.171 (-0.426, 0.085) |
| Wealth index | -1197.571 (-2403.669, 8.526) | -76.380 (-440.318, 287.558) | -510.270 (-1604.946, 584.406) |
| Amount of outstanding loan in the post-treatment period | 0.286 (0.214, 0.358)*** | 0.498 (0.425, 0.571)*** | 0.239 (0.123, 0.355)*** |
| Age | 138.945 (-12.194, 265.696)* | 44.328 (3.183, 85.472)* | -20.602 (-138.708, 97.504) |
| Gender (male) | -1171.601 (-4846.338, 2503.137) | -1037.562 (-2084.591, 9.467) | -2218.383 (-5610.626, 1173.859) |
| Adjusted $R^2$ | 0.236 | 0.412 | 0.145 |

Note: CI–confidence interval

*** indicates p value less than 0.001

* indicates p value less than 0.05

condition of the TB patients beyond treatment completion. Another unique feature of the study is that the study covered wide range of patients: general population from both urban and rural areas as well as high-risk groups such as patients in tea garden areas and urban slum areas sampled from 52 TB Units and 182 tea gardens across 9 districts.

The study found that financial hardship that started from pre-treatment phase continued in the post-treatment period. Unemployment rate among the patients were higher in post-treatment phase as compared to pre-treatment phase indicating that they were unable to return to the pre-TB condition even after one-year post-treatment. Patients from higher income groups gradually moved to lower income groups during IP, situation improved during CP but could not completely revive during post-treatment. Further, patients used several coping strategies (borrowing/selling belongings) and faced other economic consequences even during post-treatment period. Therefore, not only the physical morbidity continues after treatment completion, economic impact of TB also persists way beyond treatment completion.

The present study findings corroborate with another study conducted in Malawi which reported that substantial financial hardship experienced during TB treatment extended to 12 months post-treatment completion [9]. The study found that although the proportion of participants working in the post-treatment period increased but it did not reach the baseline and persistent dissaving was widely observed. The study also reported ongoing respiratory morbidity after treatment completion like present study findings. 30%-34% participants among all groups in the present study reported having at least one symptom like pulmonary TB. It is therefore obvious that there will be additional treatment cost even during post-treatment period as observed in the present study.

The Malawi study noted that the reasons of lower recovery after treatment completion were ongoing financial insecurity from initial TB disease, reduced social capital, TB related stigma and ongoing respiratory morbidity [9]. The present study also found statistically significant association between financial hardship in the post-treatment period and higher direct cost of initial TB treatment and treatment cost for post-treatment sequalae. This clearly emphasized the need of reducing the treatment cost of TB in the country. Further, there was no significant association between post-treatment financial distress and socio-economic and demographic characteristics implying that financial insecurity continued for all categories of patients.

Costs of initial TB treatment reported in this study (ranging from US$341 –US$386) were substantially higher than the costs estimated in a recent literature review (US$235 at 2018 prices) [16]. The reason of high cost could be because the present study covered patients from different groups and scattered across many TB Units while most of the previous studies focused only in one district covering similar population group and locality. Indirect cost was the major contributor of total cost for patients in tea garden areas because of continuation of directly observed treatment (DOT) in most tea gardens and high hospitalization rate among the patients. On the other hand, direct cost was the major contributor for other two groups and buying nutritional food was the major expense for them during treatment period, a finding like other studies [16, 28, 29]. The Nikshay Poshan Yojana [30] introduced by the government of India in 2018 to provide nutritional support to the TB patients will be helpful in this regard if all TB patients receive the benefit on time. However, only 26%-39% study participants reported receiving the benefit during CP interviews while 9%-13% mentioned that they did not check or unaware about the receipt of the benefit in their bank accounts. The low percentage of receipt of benefit could be because of implementation challenges at its early phase, however, a better implementation in future may be helpful in reducing the costs on nutritional supplements during treatment phase. Further, patients in remote areas (especially patients in tea garden areas) found it difficult to visit banks for withdrawing the benefits, hence, any alternative method to reach the patients at the remotest corners of the country needs consideration.

Proportion of households faced catastrophic cost in the present study (35%-40%) was also significantly higher than reported in the recent literature review (7%-32%) using the same method of catastrophic cost calculation. The reason could be the same as mentioned earlier that the present study covered a wide range of TB patients as compared other studies. The cost burden for urban slum dwellers and tea garden areas was highest among the richest while one recent study using national survey data found that the burden of catastrophe for hospitalization was higher among the poor quintiles [31]. Even though the cost burden was higher among the richest, the burden of dissaving was the highest among the poorest quintile for all group of patients both during treatment and post-treatment period. This implies that even for a lower treatment cost, the poorest quintile had to use various coping methods to cover their treatment costs.

The study has few limitations. First, this study is a part of an ongoing study where we aim to interview the representative sample of 1536 DS-TB patients from three groups. As the present study covered 829 participants for whom one-year post-treatment follow up interviews were completed, the estimates are not as precise as expected. Hence, results should be interpreted with caution. However, because of clear dearth of data on post-treatment experience of TB patients in India, we decided to present an interim analysis of the available data. Second, during the follow-up interviews of the study participants, COVID-19 nationwide lockdown was ongoing and hence, it had impact on the study participants' income and employment [24]. Keeping this in mind, we added few questions to separate out impact of COVID-19 from impact of TB. For example, to understand the employment status and income in the CP,

participants were asked to report employment status and income just before lockdown started to understand only the impact on TB [24]. Finally, participants were asked to report treatment costs, health seeking behaviour, income retrospectively which were subject to some recall bias, however, biases were minimized by verifying medical records wherever available.

The study findings have implications on TB programme and policies. Major reason of continued financial hardship during the post-treatment period was high cost associated with TB treatment, treatment cost associated with post-treatment sequalae, unemployment, and reduced income. To reduce costs associated with TB treatment, post-treatment sequalae and related economic consequences, the delay from symptom to treatment initiation needs to be reduced as costs incurred in pre-treatment phase contributed a significant proportion of total TB treatment cost. Awareness generation around symptoms of TB and facilities available for free diagnosis and treatment will play an important role in this context. During COVID-19 pandemic, massive awareness campaigns were conducted in the country–similar types of initiatives may be planned along with several other ongoing activities such as active case findings, private sector engagements to achieve the target of eliminating this centuries old disease.

TB care should be designed in such a way so that it does not disrupt the livelihood of the patients and therefore, job security for the TB patients should be considered. A policy priority must continue to protect TB patients from the economic consequences of the disease by introducing paid sick leave, additional food support, better management of direct benefit transfer and improving coverage through medical insurances.

The high rate of catastrophic cost for all groups of patients clearly indicates that India has a long way to go to achieve the zero catastrophic cost target as envisioned in the END TB Strategy. Further, burden of dissaving, a potential proxy of catastrophe, was the highest among the poorest quintile which highlighted that the disease had devastating effect on the financial security of these households. This further emphasized the need of introduction of the above-mentioned risk protection measures for the TB patients in India.

## Supporting information

**S1 Table. States stratified based on levels of development and regions.**
(DOCX)

**S2 Table. Descriptive statistics of explanatory variables in regression analysis.**
(DOCX)

**S1 Fig. Fitted line plots with amount of outstanding loan as dependent variable (general population, N = 220).**
(DOCX)

**S2 Fig. Fitted line plots with amount of borrowing / selling as dependent variable (general population, N = 220).**
(DOCX)

**S3 Fig. Fitted line plots with amount of outstanding loan as dependent variable (tea garden families, N = 379).**
(DOCX)

**S4 Fig. Fitted line plots with amount of borrowing / selling as dependent variable (tea garden families, N = 379).**
(DOCX)

**S5 Fig. Fitted line plots with amount of outstanding loan as dependent variable (urban slum dwellers, N = 104).**
(DOCX)

**S6 Fig. Fitted line plots with amount of borrowing / selling as dependent variable (urban slum dwellers, N = 105).**
(DOCX)

**S1 Data. Complete dataset.**
(ZIP)

## Acknowledgments

We thank the State and District TB Officers and Senior Treatment Supervisors of the study states and districts for their support and co-operation during data collection. We extend our deepest thanks to the data collection team for their sincere efforts. Finally, we are grateful to all TB patients who participated in this study for their time with us despite being sick.

## Author Contributions

**Conceptualization:** Susmita Chatterjee, Anna Vassall.

**Data curation:** Susmita Chatterjee, Palash Das.

**Formal analysis:** Susmita Chatterjee, Palash Das, Aaron Shikhule.

**Funding acquisition:** Susmita Chatterjee.

**Methodology:** Susmita Chatterjee, Anna Vassall.

**Project administration:** Susmita Chatterjee, Palash Das.

**Supervision:** Susmita Chatterjee, Palash Das, Radha Munje, Anna Vassall.

**Validation:** Susmita Chatterjee, Palash Das, Aaron Shikhule.

**Visualization:** Susmita Chatterjee.

**Writing – original draft:** Susmita Chatterjee.

**Writing – review & editing:** Susmita Chatterjee, Palash Das, Aaron Shikhule, Radha Munje, Anna Vassall.

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
