## [Decision Letter · Decision Letter 0]

25 Oct 2022

PGPH-D-22-01297

Journey of the tuberculosis patients in India from onset of symptom till one-year post-treatment

Dear Dr. Chatterjee,

Thank you for submitting your manuscript to PLOS Global Public Health. After careful consideration, we feel that it has merit but does not fully meet PLOS Global Public Health’s publication criteria as it currently stands. Therefore, we invite you to submit a revised version of the manuscript that addresses the points raised during the review process.

We look forward to receiving your revised manuscript.

Kind regards,

Alice Zwerling, PhD

Academic Editor

Journal Requirements:

2. In the online submission form, you indicated that "The datasets generated and analysed during the current study are part of an ongoing study and are not publicly available due to the agreement between the study states and the principal investigator. However, these will be available from the corresponding author on reasonable request at the end of the main study.". All PLOS journals now require all data underlying the findings described in their manuscript to be freely available to other researchers, either 1. In a public repository, 2. Within the manuscript itself, or 3. Uploaded as supplementary information.

Additional Editor Comments (if provided):

Please consider all reviewers comments carefully. Several major areas of improvement and clarification have been requested and will be required before this paper can be re-assessed for publication.

Reviewers' comments:

Reviewer's Responses to Questions

**Comments to the Author**

1. Does this manuscript meet PLOS Global Public Health’s publication criteria? Is the manuscript technically sound, and do the data support the conclusions? The manuscript must describe methodologically and ethically rigorous research with conclusions that are appropriately drawn based on the data presented.

Reviewer #1: Partly

Reviewer #2: Yes

Reviewer #3: Yes

Reviewer #4: Partly

2. Has the statistical analysis been performed appropriately and rigorously?

Reviewer #1: No

Reviewer #2: Yes

Reviewer #3: No

Reviewer #4: Yes

3. Have the authors made all data underlying the findings in their manuscript fully available (please refer to the Data Availability Statement at the start of the manuscript PDF file)?

Reviewer #1: Yes

Reviewer #2: Yes

Reviewer #3: No

Reviewer #4: Yes

4. Is the manuscript presented in an intelligible fashion and written in standard English?

Reviewer #1: Yes

Reviewer #2: Yes

Reviewer #3: No

Reviewer #4: Yes

5. Review Comments to the Author

Reviewer #1: A very relevant paper. The economic burden of Non-Communicable Diseases (NCDs) critical issue that needs to be studied more and more. The paper is well structured. However, the authors need to work on the following:

The two outcome variables: outstanding loan amount in the post treatment period and the amount of borrowing and sale/mortgage of belongings in the post-treatment period are basically one element of copying mechanisms: Borrowing. Why have you not used other dimensions of copying mechanisms? Why only focus on borrowing indicator while there’s also dissaving (use of savings to finance health care) and sale of property especially, those tied to livelihoods? It is not clear how you combined the amount of borrowing and sale/mortgage of belongings to have one qoutcome variable.

Besides, outstanding loan amount and amount of borrowing…what’s the difference between the two? What proof do you have that the outstanding loan amount was solely to finance health care?...TB treatment? This would really be a strong assumption. You are now running into estimation issues here…

Also, binary regression models such as probit and logit would have been a better fit here in my opinion than linear regression models. This was to be determined at the point of developing data collection tools though.

What is the sample size in this study?

Reviewer #2: Hello,

The manuscript entitled ‘Journey of the tuberculosis patients in India from onset of symptom till one-year post-treatment’ presents data on the during treatment and post-treatment economic impacts on the tuberculosis patients in India. The research work is well conducted with scientific rigor. Overall, the manuscript is well written and provides evidence for policy implementation to reduce healthcare costs. A few minor suggestions for your considerations are listed below:

1. Line 110: It will be helpful to provide clear definitions for the two high risk groups included in the manuscript for readers understanding.

2. Line 119 to 121: The authors provide the sample size calculations in the supplementary; however, it will be helpful to provide the power calculations in the methods section of the main body of the manuscript, rather than supplementary section.

3. Line 137 to 145: Add a sub header ‘Impact of COVID 19 on the present study’ or present this impact in the supplement.

4. Line 152-153: Provide a source or prevailing wage rate used for the costs calculations.

5. Line 194: Add table sub headers for the stratification categories for ‘general population’, ‘tea garden families’, and ‘urban families.’

6. Page 21, results section: The authors have used a rsquared method to assess the goodness of fit measure; it would be helpful to provide the fitted line plot or prediction intervals in the supplement. Also, did the authors run a sensitivity analysis? If yes, provide information.

7. Page 27, figure 1A: It is hard for the reader to clearly read the label in the graph. I hope the graphs in the final article are higher in resolution.

I hope these comments are useful.

Reviewer #3: An important topic of out-of-pocket expenditure on treatment of TB over one-year period in three types of households are covered. There are several methodological problems in the survey because the conceptual framework on cost of TB treatment has not been described meticulously. Both pre-treatment and Post-treatment phase the recall period is not defined which affect the types of cost and the ir magnitude; e.g. longer the recall period in the post-treatment phase would lead to higher accumulation of costs. They need to do a rigorous exercise by calculating the number of days the cost details for each household in four phases; pre-treatment, IP, CP and Post-treatments and then relate to the variation in costs. No specific reason is provide to distribute equally the sample between three groups and in the paper they have provided a very highly skewed distribution of cases between these three groups. Some of the variables (also dependent variables) included in the analysis are not very well defined e.g. wealth index, household income reference period, amount of borrowing (whether outstanding at the time of re-visit during post-treatment phase, and its recall period). Recall period for hospitalisation cases has not been conceptualised and defined. At several places statistics are presented without any relative comparison (If one reads some statistics on cost or DALYs one needs to compare with similar disease to see the difference otherwise these statistics are meaningless). There are several publications on cost of TB in India using National Sample Survey dataset and authors have not touched-upon this neither in the literature search nor in discussion sections. Some of the Table titles need proper headings. One can't understand Table 2 becauses each group have two columns without any headings. Most cross-tabulations required statistical testing for differences in three groups. Results interpretation and discussion need re-writing and should be very analytical with policy implications.

Reviewer #4: This is a very interesting study that estimates patient costs for TB patients in India. While the analysis is neat, it could be extended to increase impact. Some comments for the authors' consideration.

Abstract, lines 44-45- It would be good to name the Indian states in which the research was conducted.

Introduction - this section clearly explains that TB patients bear high costs, and that these are catastrophic but doesn't state why it is important to understand the full extent of the costs, and how this evidence could be relevant for health financing policy solutions or public health/ health systems policy. This weakens the motivation of the paper to some extent.

Methods

- The section on study design requires further details in the main text, especially which states was this research conducted in, along with some background information to set the frame of reference for the reader.

- Lines 127-132 summarise the data collected, which appears to be medical costs only. This should be clearly stated in this section.

- Non medical costs were not included, understandably because of the complexities introduced by the pandemic. This should be mentioned in the limitations.

- Were these costs regressive? I would imagine that they would be. It would be good to add this analysis to extend the impact of the results.

- Was it not possible to calculate catastrophic expenditure using the Feb 2020 estimates of incomes, with a ceterus paribus assumption, that could have indicated the extent of the impact that costs could have on households?

Discussion: The regressively of financing/ coping mechanisms would make this a more compelling section of the manuscript (page 18. Sorry, I can't refer to the line numbers as these disappear after page 13).

6. PLOS authors have the option to publish the peer review history of their article (what does this mean?). If published, this will include your full peer review and any attached files.

**Do you want your identity to be public for this peer review?** For information about this choice, including consent withdrawal, please see our Privacy Policy.

Reviewer #1: No

Reviewer #2: No

Reviewer #3: **Yes: **Anil Gumber

Reviewer #4: No

---

## [Decision Letter · Decision Letter 1]

17 Jan 2023

Journey of the tuberculosis patients in India from onset of symptom till one-year post-treatment

PGPH-D-22-01297R1

Dear Dr. Chatterjee,

We are pleased to inform you that your manuscript 'Journey of the tuberculosis patients in India from onset of symptom till one-year post-treatment' has been provisionally accepted for publication in PLOS Global Public Health.

Best regards,

Alice Zwerling, PhD

Academic Editor

Reviewer Comments (if any, and for reference):

Reviewer's Responses to Questions

**Comments to the Author**

1. If the authors have adequately addressed your comments raised in a previous round of review and you feel that this manuscript is now acceptable for publication, you may indicate that here to bypass the “Comments to the Author” section, enter your conflict of interest statement in the “Confidential to Editor” section, and submit your "Accept" recommendation.

Reviewer #1: (No Response)

Reviewer #4: All comments have been addressed

2. Does this manuscript meet PLOS Global Public Health’s publication criteria? Is the manuscript technically sound, and do the data support the conclusions? The manuscript must describe methodologically and ethically rigorous research with conclusions that are appropriately drawn based on the data presented.

Reviewer #1: Yes

Reviewer #4: Yes

3. Has the statistical analysis been performed appropriately and rigorously?

Reviewer #1: No

Reviewer #4: Yes

4. Have the authors made all data underlying the findings in their manuscript fully available (please refer to the Data Availability Statement at the start of the manuscript PDF file)?

Reviewer #1: Yes

Reviewer #4: Yes

5. Is the manuscript presented in an intelligible fashion and written in standard English?

Reviewer #1: Yes

Reviewer #4: Yes

6. Review Comments to the Author

Reviewer #1: The authors have tried to respond to a number of issues raised. However, I don’t quite find any plausible reason why they combined amount of borrowing and the amounts from sale/mortgage of personal belongings. These are both elements of distress financing and need not to be combined. Can the author’s run regression models separately having ‘borrowing’ and ‘sale of property.’

The variable in question is stated as: Dependent variable: Amount of borrowing and amount received from sale / mortgage of personal belongings in the post-treatment period.

Reviewer #4: All comments have been addressed. Thank you.

7. PLOS authors have the option to publish the peer review history of their article (what does this mean?). If published, this will include your full peer review and any attached files.

**Do you want your identity to be public for this peer review?** For information about this choice, including consent withdrawal, please see our Privacy Policy.

Reviewer #1: No

Reviewer #4: No
